# Reducing Collision Checking for Sampling-Based Motion Planning Using Graph Neural Networks

**Chenning Yu**
Computer Science and Engineering
UC San Diego
chy010@ucsd.edu

**Sicun Gao**
Computer Science and Engineering
UC San Diego
sicung@ucsd.edu

## Abstract

Sampling-based motion planning is a popular approach in robotics for finding paths in continuous configuration spaces. Checking collision with obstacles is the major computational bottleneck in this process. We propose new learning-based methods for reducing collision checking to accelerate motion planning by training graph neural networks (GNNs) that perform path exploration and path smoothing. Given random geometric graphs (RGGs) generated from batch sampling, the path exploration component iteratively predicts collision-free edges to prioritize their exploration. The path smoothing component then optimizes paths obtained from the exploration stage. The methods benefit from the ability of GNNs of capturing geometric patterns from RGGs through batch sampling and generalize better to unseen environments. Experimental results show that the learned components can significantly reduce collision checking and improve overall planning efficiency in challenging high-dimensional motion planning tasks.

## 1 Introduction

Sampling-based planning is a popular approach to high-dimensional continuous motion planning in robotics [36, 14, 31, 27, 19, 53, 35]. The idea is to iteratively sample configurations of the robots and construct one or multiple exploration trees to probe the free space, such that the start and goal states are eventually connected by some collision-free path through the sampled states, ideally with path cost minimized. This motion planning problem is hard, theoretically PSPACE-complete [48], and existing algorithms are challenged when planning motions of robots with a few degrees of freedom [34, 15, 1, 39, 9, 15]. In particular, the planning algorithms need to repeatedly check whether an edge connecting two sample states is feasible, i.e., that no state along the edge collides with any obstacle. This *collision checking* operation is the major computational bottleneck in the planning process and by itself NP-hard in general [28, 7]. For instance, consider the 7D Kuka arm planning problem in the environment shown in Figure 1. The leading sampling-based planning algorithm BIT* [19] spends about 28.6 seconds to find a complete motion plan for the robot, in which 20.2s (70.6% of time) is spent on collision checking. In comparison, it only takes 0.06s (0.2% of time) for sampling all the probing states needed for constructing random graphs for completing the search.

Learning-based approaches have become popular for accelerating motion planning. Many recent approaches learn patterns of the configuration spaces to improve the sampling of the probing states, typically through reinforcement learning or imitation learning [29, 25, 65, 47, 8]. For instance, Ichter et al. [25] and motion planning networks [47] apply imitation learning on collected demonstrations to bias the sampling process. The NEXT algorithm [8] provides a state-of-the-art design for embedding high-dimensional continuous state spaces into low-dimensional representations, while balancing exploration and exploitation in the sampling process. It has demonstrated clear benefits of using learning-based components to reduce samples and accelerate planning. However, we believe two

35th Conference on Neural Information Processing Systems (NeurIPS 2021).

aspects in NEXT can be improved, if we shift the focus from reducing *sample complexity* to reducing *collision checking*. First, instead of taking the grid-based encoding of the entire workspace as input, we can use the graphs formed by batches of samples from the free space to better capture the geometric patterns of an environment. Second, having access to the entire graph formed by samples allows us to better prioritize edge exploration and collision checking based on relatively global patterns, and avoid getting stuck in local regions. In short, with reasonably relaxed budget of samples taken uniformly from the space, we can better exploit global patterns to reduce the more expensive collision checking operations instead. Figure 1 shows a typical example of how the trade-off benefits overall planning, and more thorough comparisons are provided in the experimental results section.

We design two novel learning-based components that utilize Graph Neural Networks (GNNs) to accelerate the search for collision-free paths in batch sampling-based motion planning algorithms. The first component is the GNN Path Explorer, which is trained to find collision-free paths given the environment configuration and a random geometric graph (RGG) formed by probing samples. The second component is the GNN Path Smoother, which learns to optimize the path obtained from the explorer. In both models, we rely on the expressiveness and permutation invariance of GNNs as well as attention mechanisms to identify geometric patterns in the RGGs formed by samples, and accelerate combinatorial search.

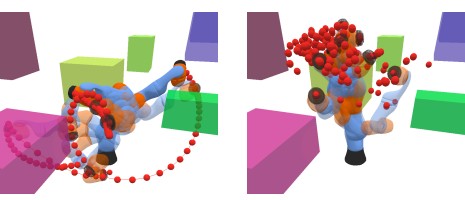

Figure 1: Performance on 7D Kuka arm. Left: Trajectory generated by the proposed GNN-based approach. Right: NEXT getting stuck at a local region. Both methods were trained on the same training set.

The proposed learning-based components can accelerate batch sampling-based planning algorithms without compromising probabilistic completeness properties. The methods achieve higher success rate, much lower rate of collision checking, and accelerate the overall planning compared to state-of-the-art methods. We evaluate the proposed approach in a variety of planning environments from 2D mazes to 14D dual KUKA arms. Experiments show that our method significantly reduces collision checking, improves the overall planning efficiency, and scales well to high-dimensional tasks.

The paper is organized as follows. We review related work and preliminaries in Section 2 and 3. We describe the detailed design of the GNN architectures in Section 4, followed by the training of GNN explorer and smoother in Section 5. We discuss experimental results in Section 6.

## 2   Related Work

**Learning-based Motion Planning.** Learning-based approaches typically consider motion planning as a sequential decision-making problem, which could be naturally tackled with reinforcement learning or imitation learning. With model-based reinforcement learning, DDPG-MP [29] integrates the known dynamic of robots and trains a policy network. Strudel et al. [54] improves the obstacle encoding with the position and normal vectors. OracleNet [2] learns via oracle imitation and encodes the trajectory history by an LSTM [23]. Other than predicting nodes which are expected to greedily form a trajectory, various approaches have been designed to first learn to sample vertices, then utilize sampling-based motion planning for further path exploration through these samples. L2RRT [24] first embeds high-dimensional configuration into low-dimensional representation, then performs RRT [36] on top of that. Ichter et al. [25] uses conditional VAE to sample nodes. Zhang et al. [65] learns a rejection sampling distribution. Madaan [40] encodes the explored tree with an RNN. Motion Planning Networks [47] utilizes the dropout-based stochastic auto-encoder for biased sampling. NEXT [8] projects the high-dimensional planning spaces into low-dimensional embedding discrete spaces, and further applies Gated Path Planning Networks [37] to predict the samples.

Existing learning-based approaches have considered improving *collision detection*. Fastron [13] and ClearanceNet [6] learn function approximators as a proxy to collision detection, which is disparate from our focus on reducing the steps that are needed to the collision checker, and can be improved further potentially if combined together. Another recent line of work focuses on learning to explore edges given fixed graph. Value Iteration Networks [55] and Gated Path Planning Networks [37] apply convolutional neural networks (CNN) on discrete maps, then predict the policy with a weighted attention sum over neighborhoods. Generalized Value Iteration Networks [44] and Velickovic et al.

[60] extend this approach for nontrivial graph by replacing CNN with GNN. However, the construction of such graphs requires ground-truth collision status for every edge on the graph at inference time.

It should be noted that other than sampling-based approaches, trajectory optimization [30, 68, 57, 51, 67], and motion primitives [43, 66] are standard choices for more structured problems such as for autonomous cars and UAVs, while sampling-based methods are important for navigating high-dimensional cluttered spaces such as for manipulators and rescue robots.

**Graph Neural Networks for Motion Planning.** Graph neural networks are permutation invariant to the orders of nodes on graph, which become a natural choice for learning patterns in graph problems. They have been successfully applied in robotics applications such as decentralized control [38]. For sampling-based motion planning, Khan et al. [33] utilizes GNN to identify critical samples. We focus on the different aspect of collision checking with given random geometric graphs, and can be combined with existing techniques without affecting probabilistic completeness. More broadly, GNNs have been used for learning patterns in general graph-structured problems, e.g. graph-based exploration [12, 52], combinatorial optimization [32, 16, 5], neural algorithm execution [60, 61, 63, 59]. Other than to use GNN for high-dimensional planning, several works propose to first learn neural metrics, then build explicit graphs upon the learned metric which is used later to search the path [50, 18, 17, 64]. While sharing similar interests, our work specifically focuses on how to reduce the collision checking steps for sampling-based motion planning.

**Informed Sampling for Motion Planning.** A main focus in motion planning is on developing problem-independent heuristic functions for prioritizing the samples or edges to explore. Approaches include Randomized A* [14], Fast Marching Trees (FMT*) [27], Sampling-based A* (SBA*) [45], Batch Informed Trees (BIT*) [19]. These methods are orthogonal to our learning-based approach, which can further exploit the problem distribution and recognize patterns through offline training to improve efficiency. Recent work in motion planning has made significant progress in reducing collision checking through batch sampling and incremental search, such as in BIT* [19] and AIT* [53]. The idea is to start with batches of probing random samples in the free space, and focus on reducing collision checking to edges that are likely on good paths to the goal, which also inspires our work.

**Lazy Motion Planning.** Lazy motion planning also focuses on reducing collision checking, typically with hand-crafted heuristics or features. LazyPRM and LazySP [4, 21] construct an RGG first and check the edge lazily only if that edge is on the global shortest path to the goal. Instead of calculating a complete shortest path, LWA* uses one-step lookahead to prioritize certain edges [11]. LRA* interleaves between LazySP and LWA*, with a predefined horizon to lookahead [41]. These approaches use hand-crafted heuristics and do not utilize data-dependent information from specific tasks. Recently, GLS and StrOLL [42, 3] leverage experiences to learn to select the edge to check with either fixed graphs or hand-crafted features. Our GNN-based approach proposes the use of new representations for the learning-based components, with the goal of directly recognizing patterns using samples from the configuration space.

## 3 Preliminaries

**Motion Planning.** We focus on motion planning in continuous spaces, where the *configuration space* is $C \subseteq \mathbb{R}^n$. The configuration space includes all degrees of freedom of a robot (e.g. all joints of a robotic arm) and is different from the *workspace* where the robot physically resides in, which is at most 3-dimensional. For planning problem on a graph $G = \langle V, E \rangle$, we denote the start vertex and goal vertex as $v_s, v_g \in C$. A path from $v_s$ to $v_g$ is a finite set of edges $\pi = \{e_i : (v_i, v_i')\}_{i \in [0,k]}$ such that $v_0 = v_s$, $v_k' = v_g$, and $v_i' = v_{i+1}$ for all $i \in [0, k-1]$. An environment for a motion planning problem consists of a set of obstacles $C_{obs} \subseteq C$ and free space $C_{free} = C \setminus C_{obs}$. Note that $C_{obs}$ is the projection of 3D objects in the higher-dimensional configuration space, and typically has complex geometric structures that can not be efficiently represented. A sample state $v \in C$ in the configuration space is free if $v \in C_{free}$, i.e., it is not contained in any obstacle. An edge connecting two samples is free if $e \subseteq C_{free}$. Namely, for every point $v$ on the edge $e$, $v \in C_{free}$. A path $\pi$ is free if all its edges are free. A random geometric graph (RGG) is a r-disc or k-nearest-neighbor (k-NN) graph $G$ [20, 62], where nodes are randomly sampled from the free space $C_{free}$. In this paper we consider the RGG as a k-NN graph. Given a random geometric graph $G$ and a pair of start and goal configuration $(v_s, v_g)$, the goal of agent is to find a free path $\pi$ from $v_s$ to $v_g$. Without loss of generality, we consider the cost of a path to be the total length over all edges in it.

**Graph Neural Networks and Attention.** Let $G = \langle V, E \rangle$ be a finite graph where each vertex $v_i$ is labeled by data $x_i \in \mathbb{R}^n$. A graph neural network (GNN) learns the representation $h_i$ of node $v_i$ by aggregating the information from its neighbors $\mathcal{N}(v_i)$. Given fully-connected networks $f$ and $g$, a typical GNN encodes the representation $h_i^{(k+1)}$ of the node $v_i$ after k aggregation as:

$$c_i^{(k)} = \oplus^{(k)}(\left\{ f(h_i^{(k)}, h_j^{(k)}) \mid (v_i, v_j) \in E \right\}) \text{ and } h_i^{(k+1)} = g(h_i^{(k)}, c_i^{(k)}) \tag{1}$$

where $h_i^{(1)} = x_i$ and $\oplus$ is some permutation-invariant aggregation function on sets, such as max, mean, or sum. We will also use the attention mechanism when we need to encode a varied number of obstacles as inputs. Suppose there are $n$ keys each with dimension $d_q$: $K \in \mathbb{R}^{n \times d_k}$, each key corresponding to a value $V \in \mathbb{R}^{n \times d_v}$. Given $m$ query vectors $Q \in \mathbb{R}^{m \times d_k}$, we use a typical attention function $\text{Att}(K, Q, V)$ for each query as $\text{Att}(K, Q, V) = \text{softmax}(QK^T / \sqrt{d_k})V$ [58]. The function is also permutation-invariant so the order of obstacles does not affect the output.

# 4 GNN Architecture for Path Exploration and Smoothing

## 4.1 Overall Approach

At a high level, motion planning with batch sampling typically proceeds as follows [19]. We first sample a batch of configurations in the free space, together with the start and goal states, and form a random graph (RGG) by connecting neighbor states (such as k-NN). A tree is then built in this graph from the start towards the goal via heuristic search. The tree can only contain collision-free edges, so each connection requires collision checking. When a path from start to goal is found, it is stored as a feasible plan, which can be later updated to minimize cost. After adding all collision-free edges in the current batch in the tree, a new batch will be added and the tree is further expanded. The algorithm keeps sampling batches and expanding the tree until the computation budget is reached. It then returns the best path found so far, or failure if none has been found.

We use GNN models to improve two important steps in this planning procedure: path exploration and path smoothing. The GNN path explorer finds a feasible path $\pi$ from start to goal given a randomly batch-sampled RGG, with the goal of reducing the number of edges that need to be checked for collision. The GNN path smoother then takes the path found by the explorer and attempts to produce another feasible path with less cost. In both tasks, the GNN models aim to recognize patterns and learn solutions to the combinatorial problems and save computation. In Figure 2, we illustrate the main steps for the overall algorithm. First, in (a-c), we generate an RGG composed of the vertices randomly sampled from the free space *without collision checking* on edges. This graph will provide patterns that the GNN explorer later uses to only prioritize certain edges for collision checking. In (d), the graph is the input to the GNN path explorer, which predicts the priority of the edges to explore and only the proposed edges are checked for collision. (e): We iteratively query the path explorer with collision checking to expand a tree in the free space until the goal vertex is reached, and sample new batches when no path is found in the current graph. (f): Once a path is provided by the path explorer, it becomes the input (together with the current RGG) to the GNN path smoother component, which outputs a new path that reduces path cost via local changes to the vertices in the input path.

## 4.2 GNN Architecture

We write the GNN path explorer as $\mathcal{N}_E$ and the GNN path smoother as $\mathcal{N}_S$. Both models take in a sampled random geometric graph $G = \langle V, E \rangle$. For an $n$-dimensional configuration space, each vertex $v_i \in \mathbb{R}^{n+3}$ contains an $n$-dimensional configuration component and a 3-dimensional one-hot label. There are 3 kinds of labels for the explorer: (i) the vertices in the free space, (ii) the vertices with collision, and (iii) the special goal vertex. There are also 3 kinds of labels for the smoother: (i) the vertices on the path, (ii) the vertices in the free space, and (iii) the vertices with collision.

The vertices and the edges are first embedded into a latent space with $x \in \mathbb{R}^{|V| \times d_h}, y \in \mathbb{R}^{|E| \times d_h}$, where $d_h$ is the size of the embedding. The embeddings for the GNN explorer and smoother are different, which will be discussed later in this section. Taking the vertex and edge embedding $x, y$, the GNN aggregates the local information for each vertex from the neighbors, by performing the

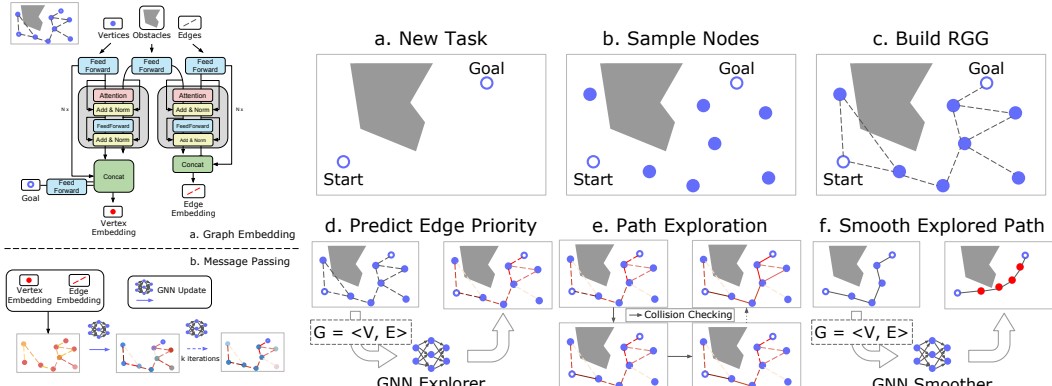

Figure 2: Left: GNN architecture shared by the path explorer and the path smoother. Right (a-f): Main steps in planning with GNNs, as explained in Section 4.1.

following operation with 2 two-layer MLPs $f_x$ and $f_y$:

$$x_i = g\left(x_i, \max\{f_x(x_j - x_i, x_j, x_i, y_l) \mid e_l : (v_i, v_j) \in E\}\right), \forall v_i \in V$$
$$y_l = \max(y_l, f_y(x_j - x_i, x_j, x_i)), \forall e_l : (v_i, v_j) \in E$$

(2)

Note that here we use $\max$ as the aggregation operator to gather the local geometric information, due to its empirical robustness to achieve the order invariance [46]. The edge information is also incorporated by adding $y_l$ as the input to $f_x$. The update function $g$ is implemented in two different ways for the GNN explorer and smoother. Specifically, $g$ equals to the $\max$ operator for the GNN explorer, and $g(m_i, x_i) = f_g(m_i) + x_i$ as the residual connection for the GNN smoother, where $f_g$ is a two-layer MLP. We choose $\max$ operator for the explorer, due to its inductive bias to imitate the value iteration, as mentioned by Velickovic et al. [60]. The residual connection is applied to the smoother, since intuitively the residual provides a direction for the improvement of each node on the path in the latent space, which fits our purpose to generate a shorter path for the smoother.

We also note that Equation 2 directly updates on the $x$ and $y$ and is a homogeneous function similar to Tang et al. [56], which allows us to self-iterate $x$ and $y$ over multiple loops without introducing redundant layers. Both the GNN explorer and smoother leverage this property. After several iterations, with two MLPs $f_\eta, f_u$, $\mathcal{N}_E$ outputs the priority $\eta = f_\eta(y)$ for each edge, and $\mathcal{N}_S$ outputs a potentially shorter path $\pi' = \{u_i, u_i'\}$, $u_i = f_u(x_i)$ for $v_i \in \pi$.

**Special design for the GNN path explorer.** The path explorer uses the embedding of the vertices of the form $x = h_x(v, v_g, (v - v_g)^2, v - v_g)$, where $h_x$ is a two-layer MLP with batch normalization [26]. Here we append the L2 distance and the difference to the goal to the vertex embedding, which serve as heuristics for the GNN to be more informed about which node is more valuable. The $y_l$ is simply computed as $y_l = h_y(v_j - v_i, v_j, v_i)$, where $h_y$ is also a two-layer MLP with batch normalization. Optionally, it is helpful for the explorer to incorporate the configuration of obstacles $O = \{o\} \in \mathbb{R}^{|\{o\}| \times 2n}$ as inputs, when embedding the vertices and edges. Since the obstacles of the environment has variable numbers, we utilize the attention mechanism here to update the $x$ and $y$, named as *obstacle encoding*, as illustrated in Figure 2. Further details are provided in the Appendix.

**Special design for GNN path smoother.** The GNN smoother embeds vertices with $x = h_x(v)$, where $h_x$ is a two-layer MLP with batch normalization. The $y_l$ is computed as $y_l = h_y(v_j - v_i, v_j, v_i)$, where $h_y$ is a two-layer MLP with batch normalization. Each time $x$ and $y$ are updated by Equation 2, the GNN smoother will output a new smoother path $\pi' = \{(u_i, u_i')\}_{i \in [0,k]}$, where $u_i = f_u(x_i), \forall v_i \in \pi$, given an MLP $f_u$. The $u_0$ and $u_k'$ are manually replaced by $v_s$ and $v_g$, to satisfy the path constraint. We assume the $\pi'$ has the same number of nodes as $\pi$. Since the GNN smoother could gain novel local geometric information with the changed vertices of the new path, we dynamically update $G = \langle V, E \rangle$, via (i) replacing those nodes labeled as path nodes in $V$ by the nodes on new path, (ii) replacing $E$ by generating a k-NN graph on the updated $V$. With the updated graph $G$, we repeat the above operation. During training, the GNN smoother outputs $\pi'$, after a random number of iterations (between 1 and 10). During evaluation, the GNN smoother outputs $\pi'$ after only one loop for each calling.

# 5    Training the Path Explorer and Smoother

Due to space limitation we provide the pseudocode for all algorithms in the Appendix.

## 5.1    GNN Explorer $\mathcal{N}_E$: Training and Inference

The path explorer constructs a tree through sampled states with the hope of reaching the goal state in a finite number of steps. We initialize the tree $\mathcal{T}_0$ with the start state $v_s$ as its root. Every edge $e_{\mathcal{T}_i}$ in the tree $\mathcal{T}_i$ exists only if $e_{\mathcal{T}_i}$ is in the free space $C_{free}$. Given an RGG $G = \langle V, E \rangle$, Our goal is to find a tree containing the goal configuration $v_g$ by adding edges from $E$ to the tree, with as few collision checks as possible. We write the edge on frontier of the tree as $E_f(\mathcal{T}) = \{(v_i, v'_i) \mid v_i \in V_{\mathcal{T}}, v'_i \notin V_{\mathcal{T}}\}$. We denote the set of edges with unknown collision status at time step $i$ as $E_i$.

**Training procedures.**    Each training problem consists of a set of obstacles $O$, start vertex $v_s$, goal vertex $v_g$, we sample a k-NN graph $G = \langle V, E \rangle$, where $V$ is the random vertices sampled from the free space combined with $\{v_s, v_g\}$. The goal is to train $\mathcal{N}_E$ to predict exploration priority $\eta \in \mathbb{R}^{|E|}$.

A straightforward way for supervision is to use the Dijkstra's algorithm to compute the shortest feasible path from $v_s$ to $v_g$, and maximize the corresponding values of $\eta$ at the edges of this path, via cross entropy loss or Bayesian ranking [49]. However, it does not provide useful guidance when the search tree deviates from the ideal optimal path at inference time. Instead, we first explore the graph using $\eta$ with $i$ steps, which forms a tree $\mathcal{T}_i$, where $i$ is a random number. The oracle provides the shortest feasible path $\pi_N$ in this tree and connects one of the nodes on $\mathcal{T}_i$ to the goal vertex $v_g$. We formulate this optimal path as $\pi_N = \{e_{N_i} : (v_{N_i}, v'_{N_i})\}_{i \in [0,k]}$, where $v_{N_0} \in V_{\mathcal{T}_i}, v'_{N_k} = v_g$. We train the explorer to imitate this oracle. Namely, the explorer will directly choose $e_{N_0} \in \pi_N$ as the next edge to explore, among all possible edges on the frontier of $\mathcal{T}_i$, i.e. $E_i \cap E_f(\mathcal{T}_i)$. We maximize the $\eta_{N_0}$ among the values of $\{\eta_i \mid e_i \in E_i \cap E_f(\mathcal{T}_i)\}$ using the following cross entropy loss:

$$L_{\mathcal{N}_E} = -\log \gamma_{N_0}, \text{ where } \gamma_k = \frac{e^{\eta_k}}{\sum_{e_j \in E_i \cap E_f(\mathcal{T}_i)} e^{\eta_j}}, \forall e_k \in E_i \cap E_f(\mathcal{T}_i) \tag{3}$$

**Inference procedures.**    Given the GNN $\mathcal{N}_E$, the current explored tree $\mathcal{T}_i$ at step $i$, the RGG $G = \langle V, E \rangle$ including $v_s$ and $v_g$, environment configuration $O$, GNN path explorer aims to maximize the probability of generating a feasible path by adding $e_i$ from $E_i$ to tree $\mathcal{T}_i$ as:

$$e_i = \underset{e_k \in E_i \cap E_f(\mathcal{T}_i)}{\arg\max} \mathcal{N}_E(\eta_k \mid V, E, O) \tag{4}$$

where $\eta_k$ is the output of $\mathcal{N}_E$ for the edge $e_k$. After $e_i$ is proposed by GNN using Equation 4, we check the collision of $e_i$. If $e_i$ is not in collision with the obstacles, we add the edge $e_i$ to the tree $\mathcal{T}_i$, and remove $e_i$ from $E_i$, i.e., $E_{\mathcal{T}_{i+1}} = E_{\mathcal{T}_i} \cup \{e_i\}$, and $E_{i+1} = E_i \setminus \{e_i\}$. If $e_i$ is in collision with obstacles, we query the path explorer for the next proposed edge using Equation 4, where $E_i$ is updated as $E_i = E_i \setminus \{e_i\}$. The loop terminates when we find a collision-free edge, or when $E_i \cap E_f(\mathcal{T}_i) = \emptyset$. When the latter happens, we re-sample another batch of samples, add new samples to vertices $V$, re-construct k-NN graph $G$, re-compute $\eta$, and continue to explore the path on this new graph with the explored nodes and edges.

The exploration GNN only proposes an ordering on the candidate edges, and all possible edges may still be collision checked in the worst case. Thus, if there exists any complete path in the RGG, the algorithm always finds it. Therefore, the proposed learning-based component does not affect the probabilistic completeness of sampling-based planning algorithms [10].

## 5.2    GNN Path Smoother $\mathcal{N}_S$: Training and Inference

The GNN $\mathcal{N}_S$ for path smoothing takes an RGG and a path $\pi$ proposed by the explorer, and aims to produce a shorter path $\pi'$. Specifically, the input is a graph $G = \langle V, E \rangle$, where $V = V_\pi \cup V_f \cup V_c$, $E = E_\pi \cup E_{fc}$. Here, $V_f$ and $V_c$ are reused as the same vertices in the GNN explorer, without introducing extra sampling complexity. $E_\pi$ is composed of those pairs of the adjacent vertices on $\pi$, and $E_{fc}$ connects each vertex in $V_\pi$ to their k-nearest neighbor in $V_f \cup V_c$. Intuitively, aggregating information from $V_f \cup V_c$ can allow GNN to identify local regions that provide promising improvement on the current path, and avoid those that may yield potential collision.

**Training procedures.** We train the GNN path smoother $\mathcal{N}_S$ by imitating a smoothing oracle $\mathcal{S}$ similar to the approach of gradient-informed path smoothing proposed by Heiden et al. [22]. To prepare the training set, we iteratively perform the following two operations on each training sample path. Given a feasible path $\pi$ predicted by $\mathcal{N}_E$, the smoothing oracle first tries to move the nodes on the path $\pi$ with perturbation within range $\epsilon$. If the new path $\pi_M$ is feasible and has cost less than $\pi$, then $\mathcal{S}$ will continue to smooth on $\pi_M$. Otherwise, $\mathcal{S}$ will continue smoothing on $\pi$ via random perturbation. After several perturbation trials, the oracle further attempts to connect pairs of nonadjacent nodes directly by a line segment. If such a segment is free of collision, then the original intermediate nodes will be moved on this linear segment. Further details are in the Appendix.

After $\mathcal{S}$ reaches maximum iteration, the oracle will return the smoothed path $\pi_M = \{w_i, w_i'\}_{i \in [0,k]}$. To generate training data, we first run our trained GNN explorer for each training problem to get the initial path $\pi$. Then the oracle path $\pi_M$ and the predicted path $\pi'$ are computed, and finally the GNN is trained via minimizing the MSE loss $L_{\mathcal{N}_S} = \frac{1}{k} \sum_{i \in [1,k]} \|\mathcal{N}_S(u_i \mid V, E) - w_i\|_2^2$.

**Inference procedures.** The GNN $\mathcal{N}_S$ for path smoothing takes an explored path $\pi : \{(v_i, v_i')\}_{i \in [0,k]}$, a sampled graph $G$, and outputs a potentially shorter path $\pi' : \{(u_i, u_i')\}_{i \in [0,k]}$ which has the same number of edges as $\pi$. It is not always guaranteed that $\pi'$ is collision-free. However, such $\pi'$ still indicates directions for shortening the path, so we can improve $\pi$ towards $\pi'$ in an incremental way. The GNN smoother will try to move every node $v_i$ towards the target position $u_i$, with a small step size $\epsilon$. For each time that all the vertices are moved with a small step, we check whether each edge still holds collision-free. If not, then the vertices on the edge will undo the movement. Otherwise, the new configuration of the vertex will replace its old configuration on $\pi$. This operation will be

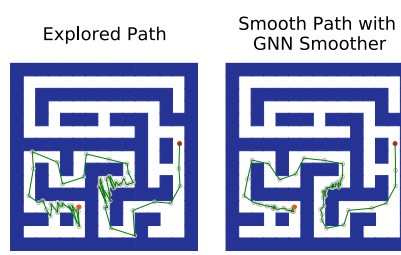

Explored Path    Smooth Path with GNN Smoother

Figure 3: GNN path smoother on the 2D maze problem. It learns to improve the explored path and achieves lower cost.

iterated over several times, until the maximum iteration is reached, or no edges on $\pi$ can be moved further. We can then repeat the process by feeding the updated $\pi$ back to the GNN $\mathcal{N}_S$ for further improvement. The intuition here is that there might still be chances to improve upon the updated $\pi$, by aggregating new information from its changed neighborhoods. This has shown empirical advantage in our experiments.

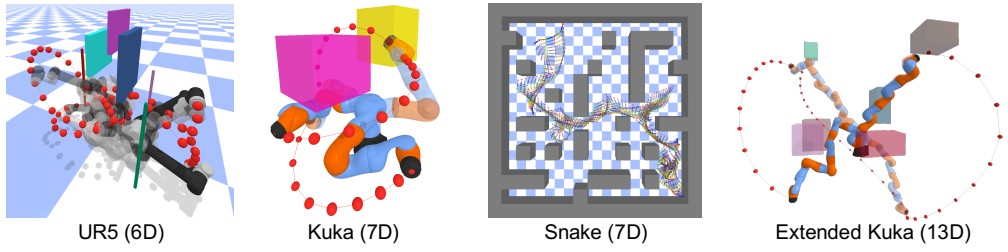

UR5 (6D)          Kuka (7D)          Snake (7D)          Extended Kuka (13D)

Figure 4: Demonstrations for some of our environments from UR5 to 13D.

# 6 Experiments

## 6.1 Overall Performance

We compare our methods with the sampling-based planning baseline RRT* [31], the state-of-the-art batch-sampling based method BIT* [19], the lazy motion planning method LazySP [21], and the state-of-the-art learning-based method NEXT [8]. NEXT has been shown in [8] to outperform competing learning-based approaches. We conduct the experiments on the following environments: (i) a 2D point-robot in 2D workspace, (ii) a 6D UR5 robot in a 3D workspace, (iii) a 7D snake robot in 2D workspace (the z-axis is fixed), (iv) a 7D KUKA arm in a 3D workspace, (v) an extended 13D KUKA arm in 3D workspace, (vi) and a pair of 7DoF KUKA arms (14 DoF) in 3D workspace.

For each environment, we randomly generate 2000 problems for training and 1000 problems for testing. Each problem contains a different set of random obstacles, and a pair of feasible $v_s$ and $v_g$. We run all experiments over 4 random seeds. The averaged results are illustrated in Figure 5. For the 2D environment, we directly take the training set provided by NEXT to train our GNN. We use two test sets for the 2D environment: "Easy2D" is the original test set used for evaluating NEXT in the original paper [8], and "Hard2D" is a new set of tests we generated by requiring the distance between the start and goal to be longer than the easy environments. The goal is to test whether the learned models can generalize to harder problems without changing the training set. Further details on the environments and hyperparameters are provided in the Appendix.

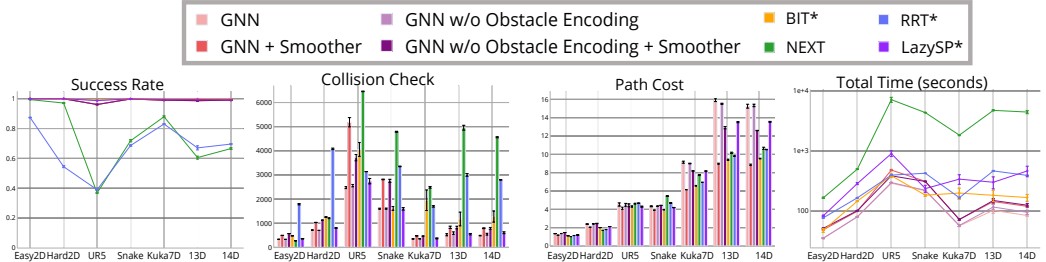

Figure 5: Comparison of performances on all environments from 2D to 14D, averaged over 4 random seeds. From left to right: (a) Success rate. (b) Collision checks. (c) Path cost. (d) Total planning time.

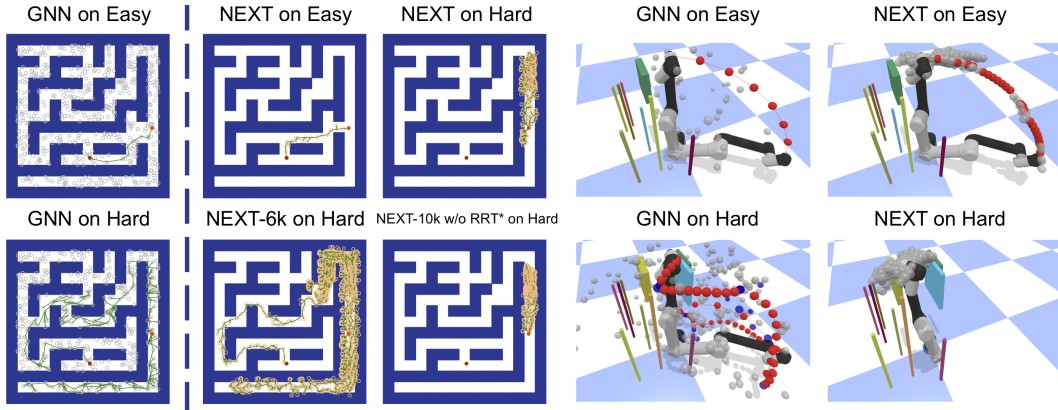

Figure 6: We test the generalization capability of GNN-based approaches and NEXT by constructing pairs of problems that have small but important difference in connectivity. The GNN models find paths on both environments quickly, while NEXT gets stuck in hard instances because of the lack of access to the graph structure provided by probing samples. The explored vertices of GNN on UR5 environment are colored in blue, and the edges on the path are colored in red.

**Success rate.** As shown in Figure 5 (a), our method finds complete paths at 100.0% problems on both 2D Easy and 2D Hard, and at 97.18%, 99.85%, 99.20%, 99.15%, and 99.15% problems from UR5 to 14D, which is comparable to handcrafted heuristics used in BIT* (100.0%, 100.0%, 99.25%, 99.85%, 99.65%, 99.92%, 99.82% on each environment). The learning-based planner NEXT performs well on easy 2D problems (99.37% success rate), but drops slightly to 97.10% on harder 2D problems, and 36.80%, 71.80%, 87.9%, 60.52%, 66.57% on environments in higher dimensions.

**Collision checking.** In Figure 5 (b), we see significant reduction of collision checking using the proposed approach, in comparison to other approaches especially in high-dimensional problems. The average number of collision checks by the GNN explorer is 336.3, 715.7, 2474.0, 1602.2, 350.5, 521.7, 487.0 on the environments, whereas BIT* needs 112%, 175%, 164%, 101%, 557%, 226%, 263% times as many collision checks as our method requires on each environments. LazySP needs 105%, 114%, 111%, 100%, 105%, 105%, 124% times as many collision checks as GNN requires on each environments. NEXT requires 270.23 checks on Easy2D and increases to 1206.1 on Hard2D. On the 14D environment, our method uses 17.4% of the collision checks as what NEXT requires.

**Path cost.** We show the average path cost over all problems where all algorithms successfully found complete paths. With the smoother and obstacle encoding, our GNN approach provides the best results from UR5 to 14D, and generates comparable results for the 2D Easy and 2D Hard, where NEXT is 1.02, 1.71, and GNN is 1.18, 2.05. We find that although the GNN explorer does not yield shorter path with obstacle encoding, these explored paths can be improved further with the smoother. The reason may be that with additional obstacle encoding, the GNN explorer tends to explore edges with less probability of collision and provides more space for the smoother to improve the paths.

**Planning time.** A common concern about learning-based methods is that their running cost due to the frequent calling of a large neural network model at inference time (as seen for the NEXT curve). We see in Figure5(d) that the wall clock time of using the GNN models is comparable to the standard heuristic-based LazySP, BIT* and RRT*, when all algorithms can find paths. The main reason is that the reduction in collision checking significantly reduces the overall time. We believe the GNNs can be further optimized to achieve faster inference as well.

In summary, experimental results show that the GNN-based approaches significantly reduce collision checking while maintaining high success rate, low path cost, and fast overall planning. The performance scales well from low-dimensional to high-dimensional problems.

## 6.2 Generalizability of Collision Reduction

A major challenge of learning-based approaches for planning is that small changes of the geometry of the environment can lead to abrupt change in the solutions, and thus lead the difficulty of generalization. In Figure 6, we provide evidence that the GNN approach can alleviate this problem because of its access to the graph structures formed by samples uniformly taken from the space. In the 2D maze environment, the top one is easy while the bottom one is much harder, although the difference is just whether a narrow corridor is present to the left of the start state. For the UR5 with pads and poles, to solve the hard one, the robot arm needs to first rotate itself around the z-axis to bypass the small pads, and then rotate it back to fit the goal configuration. These problems are especially challenging for generalizing learned results to unseen environments.

We observe that the GNN-based components can handle the transition from the easy to hard problems consistently. In both environments, the GNN components find paths quickly, with 9 and 236 edges explored for 2D maze, and with 1 and 256 edges explored for UR5. In contrast, the NEXT model trained on the same training sets, can quickly find a path in the easy problem, but gets stuck in the hard one. Since the two problems are close to each other in the input space for NEXT, it is not surprising to see this difficulty of generalization. In fact, the only case where NEXT can eventually find a path on 2D Maze after more than 6K edge exploration is when the algorithm delegates 10% operations to standard RRT*. Without delegating to RRT*, NEXT gets stuck in local regions after 10K exploration steps on 2D and 1K steps on UR5.

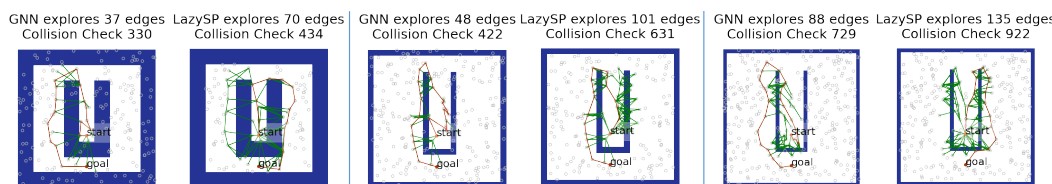

Figure 7: Comparison of performances with LazySP across different batch samples. The final path is colored red, and the other explored edges are colored green. From left to rightm we show examples of the search results with the batch size being 100, 200, and 300, respectively. We show obstacles of different thickness because the in-collision samples are also part of the inputs of the GNN planner, which can provide useful information about the topology of the environment.

## 6.3 Further Comparison with Lazy Motion Planning

We compare GNN with the lazy motion planning method LazySP [21]. Lazy approaches prioritize the collision checking on edges that are part of the shortest paths in the RGG, which is a strategy that can often see good performance especially on randomly generated graphs. However, as lazy planning uses the fixed heuristic of prioritizing certain paths, it is easy to come up with environments where

this heuristic becomes misleading. Our proposed learning-based approach, instead, uses GNN to discover patterns from the training set, and can thus be viewed as a data-dependent way of forming heuristics for reducing collision checking.

Consider U-shaped obstacles, with the start state close to the bottom of the U-shape, as shown in Figure 7. It is a standard environment that is particularly hard for the standard lazy approach, which prioritize the edges that directly connect the start and goal states, as they are close in the RGG that does not consider obstacles. Indeed, the lazy approach needs to check most of the edges that cross the obstacles before the path for getting out of the U-shape can be found. We train the GNN-based model on these environments and observe clear benefits of learning-based components in avoiding this issue. Figure 7 shows the difference between the two approaches in several examples of the 2D environment using different sizes of sample batches, where the GNN-based approach can typically save 50% of collision checking on edges compared to the lazy approach.

## 6.4 Ablation Study: Probing Samples

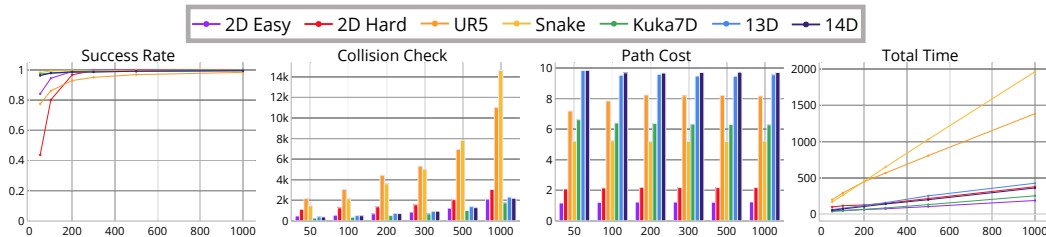

Figure 8: Comparison of performances for different probing samples from 2D to 14D problems.

We perform ablation studies of varying different parameters in our approach. *The full details are provided in the Appendix.* For instance, we use RGGs with different number of vertices from the free space, using 100, 200, 300, 500, 1000 samples, as illustrated in Figure 8. The success rate increases when there are more probing samples, which is consistent with the resolution-complete property of sampling-based planning. The path cost stays nearly the same for all settings, indicating robustness of the smoother models. The collision checks and planning time grows linearly with the probing samples. The reason is that the number of edges of k-NN RGG increments linearly with vertices, the input to the GNN grows linearly, thus the computation cost increases linearly on both CPU and GPU.

## 7 Conclusion

We presented a new learning-based approach to reducing collision checking in sampling-based motion planning. We train graph neural network (GNN) models to perform path exploration and path smoothing given the random geometric graphs (RGGs) generated from batch sampling. We rely on the ability of GNN for capturing important geometric patterns in graphs. The learned components can significantly reduce collision checking and improve overall planning efficiency in complex and high-dimensional motion planning environments.

## Acknowledgments and Disclosure of Funding

This material is based upon work supported by the United States Air Force and DARPA under Contract No. FA8750-18-C-0092, AFOSR YIP FA9550-19-1-0041, NSF Career CCF 2047034, and NSF NRI 1830399. We thank the anonymous reviewers for various important suggestions, such as for the comparison with lazy planning approaches.

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
