# Appendix to "Reducing Collision Checking for Sampling-Based Motion Planning Using Graph Neural Networks"

**Chenning Yu**
Computer Science and Engineering
UC San Diego
chy010@ucsd.edu

**Sicun Gao**
Computer Science and Engineering
UC San Diego
sicung@ucsd.edu

## 1   More Details on GNN Architectures

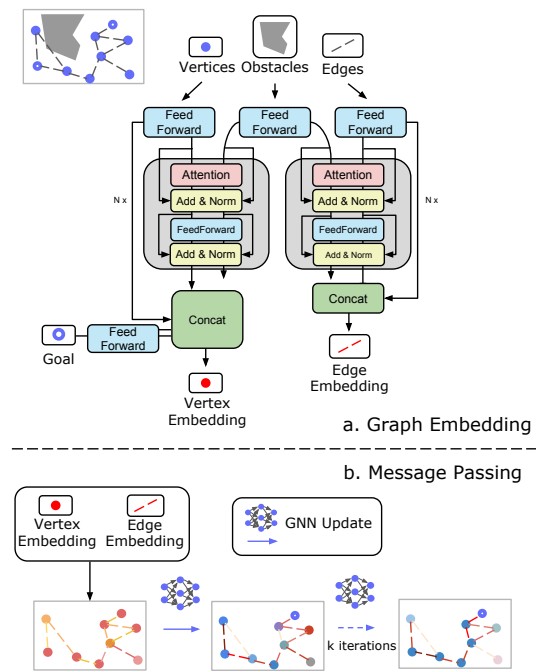

## 1.1   Obstacle Encoding

In the experiment part, we find obstacle encoding is helpful to the GNN explorer, which can optimize the explored path further with the smoother. We elaborate on the formulation of the obstacle encoding.

In this work, we consider an obstacle as a 2D or 3D box depending on the workspace, denoted as $o = (p_1, \cdots, p_n, l_1, \cdots, l_n) \in \mathbb{R}^{2n}, n \in [2, 3]$, where $p_i$ and $l_i$ are the center and length of the box along the $i$-th dimension. The environment configuration is written as $O \in \mathbb{R}^{|\{o\}| \times 2n}$, where $|\{o\}|$ is the number of the obstacles. Note that $|\{o\}|$ is a variable number, since the number of obstacles can be different for each problem.

35th Conference on Neural Information Processing Systems (NeurIPS 2021).

Given MLPs $f_{a_x}^{(i)}, f_{a_y}^{(i)}, f_{K_x}^{(i)}, f_{Q_x}^{(i)}, f_{V_x}^{(i)}, f_{K_y}^{(i)}, f_{Q_y}^{(i)}, f_{V_y}^{(i)}$, the obstacle encoding is formulated as:

$$a_x = \mathrm{LN}(x + Att(f_{K_x^{(i)}}(O), f_{Q_x^{(i)}}(x), f_{V_x^{(i)}}(O))) \text{ and } x = \mathrm{LN}(a_x + f_{a_x}^{(i)}(a_x))$$
$$a_y = \mathrm{LN}(y + Att(f_{K_y^{(i)}}(O), f_{Q_y^{(i)}}(y), f_{V_y^{(i)}}(O))) \text{ and } y = \mathrm{LN}(a_y + f_{a_y}^{(i)}(a_y))$$
$$(1)$$

where LN denotes the layer normalization [1]. This architecture follows the standard transformer block design [6].

## 1.2 Special Features

The overall special features for GNN explorer and smoother are described as follows:

**Special features in explorer architecture.** The GNN explorer embeds vertices with $x = h_x(v, v_g, (v - v_g)^2, v - v_g)$, with an MLP $h_x$. The $y_l$ is computed as $y_l = h_y(v_j - v_i, v_j, v_i)$, with an MLP $h_y$. Optionally, we utilize the obstacle encoding to update the $x$ and $y$. With Equation 1, the $x$ and $y$ will merge the information from obstacles through multiple attention blocks, which is set as 3 in our experiments.

As mentioned in the main part, the GNN explorer will update $x$ and $y$ over multiple loops. During training, we iterate $x$ and $y$ over a random number of loops between 1 and 10. Intuitively, taking random loops encourages the GNN to converge faster, which also helps propagating the gradient. During evaluation, the GNN explorer will output $x$ and $y$ after 10 loops. For loops larger than 10, significant improvement on performance is not perceived. Finally, with an MLP $f_\eta$, the GNN explorer will output $\eta = f_\eta(y)$, which will be used as the priority to explore corresponding edges.

**Special features in smoother architecture.** The GNN smoother embeds vertices with $x = h_x(v)$, with an MLP $h_x$. The $y_l$ is computed as $y_l = h_y(v_j - v_i, v_j, v_i)$, with an MLP $h_y$. Each time $x$ and $y$ are updated, the GNN smoother will output a new smoother path $\pi' = \{(u_i, u'_i)\}_{i \in [0,k]}$, where $u_i = f_u(x_i), \forall v_i \in \pi$, given an MLP $f_u$. The $u_0$ and $u'_k$ are manually replaced by $v_s$ and $v_g$, to satisfy the path constraint. We assume the new smoother path has the same number of nodes as the original path. Since the GNN smoother could gain novel local geometric information with the changed configuration of the new path, we dynamically update $G = \langle V, E \rangle$, via (i) replacing those nodes labeled as path nodes in $V$ by the nodes on new path, (ii) replacing $E$ by generating a k-NN graph on the updated $V$. With the updated graph $G$, we repeat the above operation, and subsequently get another new path, which forms a loop. By updating the graph and the new path iteratively and dynamically, the path is potentially improved to be shorter at each round by perceiving the changing local neighbors. During training, the GNN smoother outputs $\pi'$, after a random number of loops, which is between 1 and 10. During evaluation, the GNN smoother will output $\pi'$ after only one loop, but will be called 5 times in total for each smoothing tasks.

## 2 Environments and Datasets

We conduct the experiment on 6 different environments, which are described in details as follows, :

**Maze** The maze contains a 2D point robot. The datasets for training set and the test set for "Easy2D" is at `https://github.com/NeurEXT/NEXT-learning-to-plan/tree/master/algorithm` [2]. To generate the "Hard2D", we utilize the script provided by `https://github.com/RLAgent/gated-path-planning-networks` [5]. The Hard mazes are generated by controlling the obstacle density not less than 46%, and the distance from start to goal not less than 1.

**UR5** The UR5 contains a UR5 robot arm [8], which has 6 degrees of freedom. There are two sets of boxes, poles and pads, which are set to generate in two different size range. The poles and pads are randomly generated in the workspace for each problem.

**Snake** The Snake environment contains a snake robot with 5 sticks, with another 2 degrees for the end position, which means 7D in total. The mazes are the same set of 2D mazes from NEXT.

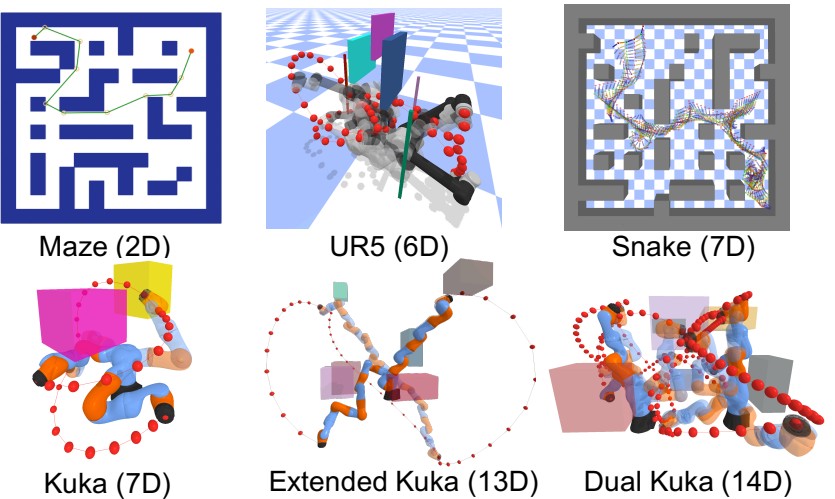



Maze (2D)   UR5 (6D)   Snake (7D)

Kuka (7D)  Extended Kuka (13D)  Dual Kuka (14D)



Figure 1: Demonstrations of all our environments.

**Kuka, Extended Kuka**  The Kuka environment contains a 7DoF Kuka arm with fixed base position. The extended Kuka environment contains an extended 13DoF kuka arm. The boxes are randomly generated in the workspace for each problem.

**Dual Kuka**  The environment contains two 7DoF KUKA arms, with 14DoF in total. Each arm need to reach the goal configuration, while required to not only avoid collision with the obstacles but also the other arm.

All the environments except the mazes are all implemented by PyBullet [3] with the MIT license. The URDF files are contained in our supplementary codes.

## 3 Tables for Overall Performance

Here we list the overall performances of all the methods on all the environments, including the averaged value with the standard deviation.

Table 1: Success rate. Our algorithm benefits from the probabilistic complete property from the RGG, which samples uniformly from free space,

|  | Easy2D | Hard2D | UR5 | Snake | Kuka7D | 13D | 14D |
|---|---|---|---|---|---|---|---|
| GNN | **1.00±0.00** | **1.00±0.00** | 0.96±0.00 | **1.00±0.00** | 0.99±0.00 | 0.99±0.00 | 0.99±0.00 |
| GNN + Smoother | **1.00±0.00** | **1.00±0.00** | 0.96±0.00 | **1.00±0.00** | 0.99±0.00 | 0.99±0.00 | 0.99±0.00 |
| GNN w/o OE | **1.00±0.00** | **1.00±0.00** | 0.96±0.00 | **1.00±0.00** | 0.99±0.00 | 0.99±0.00 | 0.99±0.00 |
| GNN w/o OE + Smoother | **1.00±0.00** | **1.00±0.00** | 0.96±0.00 | **1.00±0.00** | 0.99±0.00 | 0.99±0.00 | 0.99±0.00 |
| BIT* | **1.00±0.00** | **1.00±0.00** | **0.99±0.00** | **1.00±0.00** | **1.00±0.00** | **1.00±0.00** | **1.00±0.00** |
| NEXT | 0.99±0.00 | 0.97±0.00 | 0.37±0.00 | 0.72±0.01 | 0.88±0.01 | 0.61±0.01 | 0.67±0.00 |
| RRT* | 0.87±0.00 | 0.54±0.01 | 0.39±0.00 | 0.69±0.00 | 0.83±0.00 | 0.67±0.01 | 0.70±0.00 |
| LazySP | **1.00±0.00** | **1.00±0.00** | **0.99±0.00** | **1.00±0.00** | 0.99±0.00 | 0.99±0.00 | 0.99±0.00 |

Table 2: Collision check. GNN performs the best in most high dimensional problems.

|  | Easy2D | Hard2D | UR5 | Snake | Kuka7D | 13D | 14D |
|---|---|---|---|---|---|---|---|
| GNN | 336.25±3.71 | 715.65±6.76 | **2474.03±40.35** | 1602.16±22.66 | **350.52±8.29** | **521.70±44.58** | **486.95±12.99** |
| GNN + Smoother | 496.79±4.68 | 1029.72±8.33 | 5182.02±191.40 | 2813.75±15.01 | 477.32±9.06 | 830.95±49.06 | 791.78±14.27 |
| GNN w/o OE | 332.30±4.00 | **703.72±5.78** | 2556.63±49.91 | 1605.73±24.00 | 353.89±7.47 | 588.65±51.04 | 547.16±37.61 |
| GNN w/o OE + Smoother | 565.38±6.37 | 1126.02±9.91 | 3715.40±132.41 | 2757.88±62.64 | 466.06±6.70 | 820.70±55.97 | 789.25±36.54 |
| BIT* | 478.88±10.95 | 1253.56±15.38 | 4055.73±286.93 | 1612.22±78.85 | 1951.81±424.82 | 1175.42±287.68 | 1276.95±230.88 |
| NEXT | **270.23±13.92** | 1206.09±18.62 | 6461.13±14.31 | 4788.84±20.60 | 2488.49±33.76 | 4958.80±99.51 | 4559.99±21.92 |
| RRT* | 1785.46±27.93 | 4080.07±32.69 | 3135.36±4.03 | 3352.45±15.68 | 1698.04±28.34 | 3004.45±55.36 | 2796.99±13.89 |
| LazySP | 351.80±2.47 | 801.21±6.74 | 2742.12±113.08 | **1595.74±48.15** | 369.36±19.42 | 546.64±29.40 | 604.64±38.84 |

Table 3: Path cost. With the GNN smoother, our path cost is the lowest from UR5 to 14D.

|  | Easy2D | Hard2D | UR5 | Snake | Kuka7D | 13D | 14D |
|---|---|---|---|---|---|---|---|
| GNN | 1.34±0.01 | 2.39±0.02 | 4.54±0.17 | 4.33±0.05 | 9.15±0.11 | 15.91±0.15 | 15.26±0.21 |
| GNN + Smoother | 1.18±0.01 | 2.05±0.01 | **4.12±0.12** | **3.91±0.01** | **6.14±0.02** | **8.98±0.06** | **8.86±0.08** |
| GNN w/o OE | 1.36±0.01 | 2.41±0.03 | 4.50±0.15 | 4.35±0.03 | 9.00±0.04 | 15.52±0.06 | 15.34±0.14 |
| GNN w/o OE + Smoother | 1.46±0.01 | 2.45±0.03 | 4.45±0.15 | 4.43±0.02 | 8.18±0.06 | 12.91±0.13 | 12.59±0.03 |
| BIT* | 1.11±0.00 | 2.00±0.02 | 4.33±0.09 | 3.95±0.02 | 6.57±0.04 | 9.41±0.07 | 9.54±0.04 |
| NEXT | **1.02±0.00** | **1.71±0.01** | 4.62±0.05 | 5.45±0.04 | 7.74±0.06 | 10.17±0.07 | 10.66±0.12 |
| RRT* | 1.14±0.01 | 1.79±0.01 | 4.66±0.03 | 4.69±0.05 | 6.95±0.02 | 9.81±0.04 | 10.52±0.03 |
| LazySP | 1.20±0.01 | 2.11±0.01 | 4.30±0.06 | 4.18±0.03 | 8.16±0.05 | 13.51±0.06 | 13.54±0.06 |

Table 4: Total running time. GNN requires low time cost due to its optimization on collision checks.

|  | Easy2D | Hard2D | UR5 | Snake | Kuka7D | 13D | 14D |
|---|---|---|---|---|---|---|---|
| GNN | **35.07±0.78** | 80.57±1.16 | **290.41±3.20** | 218.83±3.41 | **56.77±1.89** | 102.67±10.48 | **84.25±3.16** |
| GNN + Smoother | 48.71±0.79 | 99.40±1.18 | 481.31±13.31 | 312.30±2.64 | 72.32±1.93 | 143.71±11.00 | 119.68±2.95 |
| GNN w/o OE | 35.43±0.66 | **80.19±1.04** | 298.18±4.25 | 221.26±2.29 | 57.62±1.87 | 116.69±11.75 | 95.19±8.36 |
| GNN w/o OE + Smoother | 51.59±0.76 | 101.40±1.17 | 386.30±9.97 | 310.42±4.39 | 72.39±1.85 | 149.86±12.19 | 125.09±8.25 |
| BIT* | 47.69±3.43 | 146.55±2.88 | 387.44±27.34 | **183.84±6.69** | 199.19±42.31 | 183.54±40.22 | 167.81±21.43 |
| NEXT | 166.46±4.81 | 499.65±4.40 | 7150.17±690.44 | 4355.78±28.44 | 1837.28±40.04 | 4750.26±88.29 | 4450.67±278.09 |
| RRT* | 77.20±1.03 | 166.06±1.35 | 396.86±0.45 | 425.87±2.46 | 166.98±2.43 | 465.80±8.58 | 392.57±2.74 |
| LazySP | 83.65±1.97 | 287.30±11.93 | 905.22±97.21 | 236.79±31.76 | 339.97±61.63 | 301.63±67.92 | 465.30±93.14 |

## 4 Breakdown of the Total Planning Time

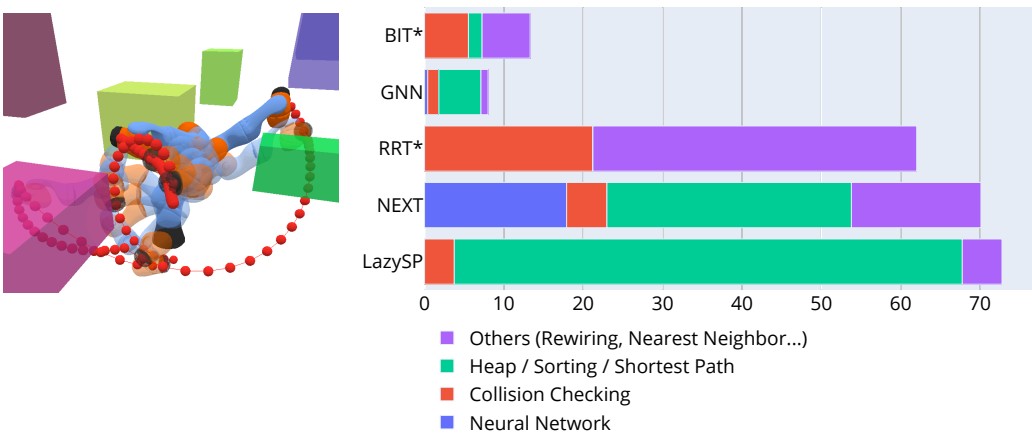

Figure 2: On the left environment, we breakdown the total planning time into various operations. The time taken by each operation is shown on the right chart.

As shown in Figure 2, we breakdown the total planning time into various operations on an example environment. We found that our method takes most of the time sorting the priority on the frontier. On contrary, BIT* and RRT* take relatively large amount of time checking for collisions. NEXT needs to recalculate the exploration bonus, and sort the candidates to explore, which takes prohibitive computation. LazySP searches for global shortest path every time an edge is in collision, which makes the search on path become the bottleneck.

## 5 Ablation Study

### 5.1 Varying Training Set Size for Explorer

We conduct further experiments to analyze the effect of the training set size. We train the GNN path explorer with the 0 (0%), 10 (0.5%), 40 (2%), 200 (10%), 400 (20%), 800 (40%), 1200 (60%), 1600 (80%), 2000 (100%) problems in this new training set. The performance of collision checks, path costs, and total time on the testing problems are demonstrated in Figure 3.

As shown, the overall performance of the GNN explorer is robust, even with 2% problems of the original training set. There are two reasons here: (i) The GNN models we propose are relatively

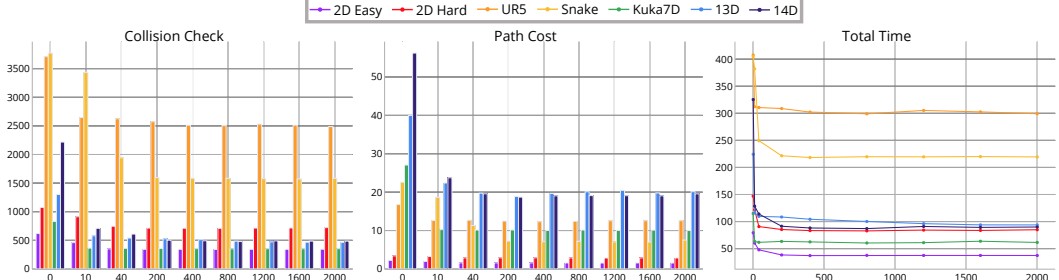

Figure 3: Ablation study on the different training set size. All the performances become stable at relatively few training set size (around 40 problems, 2% of the original training set).

lightweight in terms of the parameter numbers, which means that it is suffice to train it with small amount of data. (ii) Our GNN model does not depend on the global feature of the whole graph, as it only aggregates the information from local neighborhoods. Though each problem yields a different graph in terms of global characteristic, they can share similar local geometric patterns, which is beneficial for the efficiency of learning GNN models.

## 5.2 Feature Choices

In this experiment, we replace the vertex embedding $x = h_x(v, v_g, (v - v_g)^2, v - v_g)$ by $x = h_x(v, v_g)$, which removes the L2 distance heuristic on features. We retrain the new GNN with the same training set, and compare to the original architecture on 4 environments. As shown in Figure 4, the performances of two GNNs are close to each other from 2D to 7D environments (0.5%, 0.6%, 1.0% in terms of collision checking), and the original GNN is slightly better on 14D environment (6.8% in terms of collision checking, 0.9% in terms of path cost). The L2 distance heuristic is helpful in high dimensions, but does not have much effect, due to the complex geometry of the C-space.

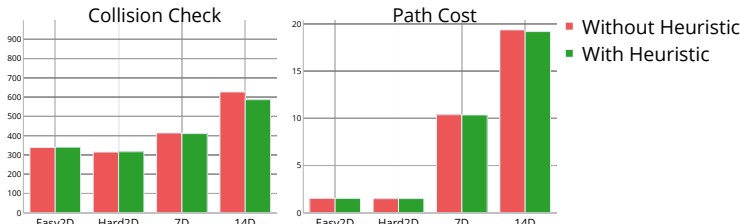

Figure 4: Comparison of performance of GNN explorers with vertex embedding $x$ as $h_x(v, v_g)$ and $h_x(v, v_g, (v - v_g)^2, v - v_g)$ respectively. Results show that the GNN explorer with heuristics perform slightly better for high-dimensional problems.

## 5.3 GNN Smoother Versus Oracle Smoother

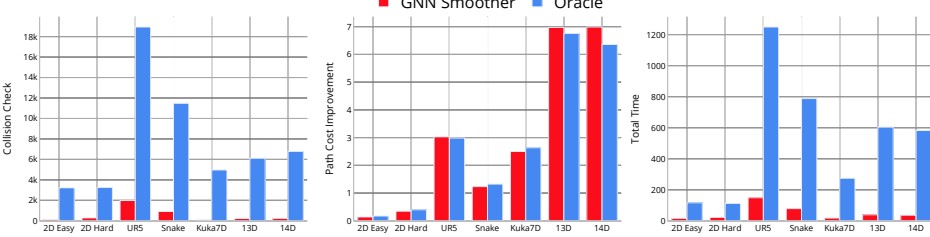

Figure 5: Comparison of performance between our GNN smoother and the oracle to trained on. Our GNN smoother learns to smooth the path with comparable improvement as the oracle, and also requires fewer collision checking steps and less total time.

In this experiment, we replace the learned GNN smoother by the oracle smoother, which is the expert that GNN smoother imitates. We compare these two smoothers, given the same path explored by

the GNN explorer on the test problems. As shown in Figure 5, our smoother requires much fewer collision checks and time, while maintaining comparable improvement on the explored path, which is contributed by the incremental way of smoothing, and the generalizablity of the GNN. More specifically, the GNN smoother requires 3.9%, 8.8%, 10.5%, 8.0%, 1.9%, 3.7%, 3.5% as many collision checks as the oracle on each environment, while maintaining 80.7%, 86.9%, 101.5%, 93.2%, 94.9%, 103.2%, 109.9% as much improvement as the oracle for each environment.

### 5.4 Varying the $k$ in k-NN

As suggested in Xue and Kumar [7], we set the $k$ in the k-NN graph as proportional to the logarithm of the number of vertices, which is formulated as $\lceil k_0 \cdot \frac{\log |V_f|}{\log 100} \rceil$. Here we test different $k_0$ for the k-NN, choosing among $\{1, 2, 4, 10, 20, 40\}$, as demonstrated in Figure 6.

It is not surprising for the success rate to increase when $k$ increases, since there are more edges in the graph, which increases the possibility to find a feasible path. The path cost also decreases with increasing $k$, since on average there might be fewer segments on a path. The collision checks and total running time first decrease then increase, since larger $k$ brings higher possibility to find a feasible path with fewer intermediate vertices, but also brings more edges to check on the frontier.

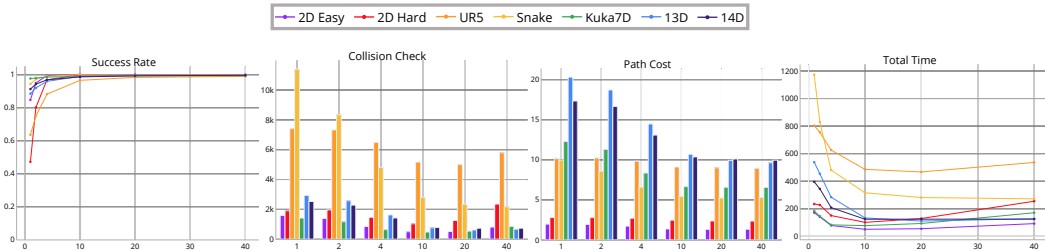

Figure 6: Ablation study on different $k_0$ for k-NN. Performances are best at $k_0 \in [10, 20]$.

### 5.5 Varying the Batch Size

In this experiment, we inspect the effect of the batch size. While constraining the maximum sampling number from free space to be 1000, we set the batch sampling size among $\{50, 100, 200, 250, 500, 1000\}$. We see that the success rate drops when the batch size increases, since the GNN explorer is given fewer opportunities to fail and re-sample. The collision checks grows with the batch size, because the graphs would contain more edges with larger batch size on average. The path cost is lower with larger batches, similar to the effect of higher $k$, due to higher possibility to find a feasible path with fewer intermediate vertices. The total time raises when the batch size goes larger, because larger batches brings denser graphs, which enlarges both CPU and GPU costs.

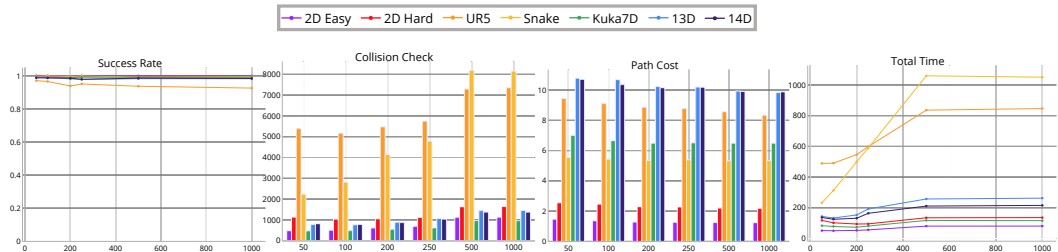

Figure 7: Ablation study on different batch size for batch sampling. Larger batches tend to yield lower success rate and path costs, while requiring more collision checks and running time.

## 6 Hyperparameters

The hyperparameters that we use are listed in the following table.

| Hyperparameters | Values |
|---|---|
| Maximum sampling number | 1000 |
| $k$ for k-NN | $\lceil 10 \cdot \frac{\log |V_f|}{\log 100} \rceil$ |
| GNN batch size | 100 |
| BIT* batch size | 100 |
| RRT*/NEXT step size on 2D | 5e-2 |
| RRT*/NEXT step size on 7D/13D/14D | 5e-1 |
| Training epoch | 20 |
| Training batch size | 8 |
| Learning rate | 1e-3 |
| Random seeds | 1234, 2341, 3412, 4123 |

# 7 Algorithms

---

**Algorithm 1:** GNNExplorer

---

    **Input:** obstacles $O$, start $v_s$, goal $v_g$, batch size $n$, node limit $T_{max}$
    Sample $n$ nodes from $C_{free}$ to $V_f$
    Sample $n$ nodes from $C_{obs}$ to $V_c$
    Initialize $G = \{V : \{v_s, v_g\} \cup V_f \cup V_c, E : \text{k-NN}(V_f) \cup \text{k-NN}(V)\}$
    Initialize $i = 0, E_0 = \text{k-NN}(V_f), V_{\mathcal{T}_0} = \{v_s\}, E_{\mathcal{T}_0} = \emptyset$
    $\eta = \mathcal{N}_E(V, E, O)$
    **repeat**
      select $e_i$ with $\eta$
      $E_i \leftarrow E_i \setminus \{e_i\}$
      **if** $e_i : (v_i, v_i') \subseteq C_{free}$ **then**
        $V_{\mathcal{T}_{i+1}} \leftarrow V_{\mathcal{T}_i} \cup \{v_i'\}$
        $E_{\mathcal{T}_{i+1}} \leftarrow E_{\mathcal{T}_i} \cup \{e_i\}$
        $E_{i+1} \leftarrow E_i$
        $E_f(\mathcal{T}_{i+1}) \leftarrow \{e_j : (v_j, v_j') \in E_{i+1} \mid v_j \in V_{\mathcal{T}_{i+1}}, v_j' \notin V_{\mathcal{T}_{i+1}}\}$
        $i \leftarrow i + 1$
        **if** $||v_i' - v_g||_2^2 \leq \delta$ **then**
          $\pi \leftarrow$ path from $v_s$ to $v_g$ on tree $\mathcal{T}_i$
          **return** $\pi$
        **end if**
      **end if**
      **if** $E_i \cap E_f(\mathcal{T}_i) == \emptyset$ **then**
        Sample $n$ nodes from $C_{free}$, add to $V_f$
        Sample $n$ nodes from $C_{obs}$, add to $V_c$
        $V \leftarrow \{v_s, v_g\} \cup V_f \cup V_c$
        $E_i \leftarrow \text{k-NN}(V_f) \setminus (E \setminus E_i)$
        $E \leftarrow \text{k-NN}(V_f) \cup \text{k-NN}(V)$
        $\eta = \mathcal{N}_E(V, E, O)$
      **end if**
    **until** $|V_f| > T_{max}$
    **return** $\emptyset$

---

The smoothing oracle that we use is similar to the approach of gradient-informed path smoothing proposed by Heiden et al. [4]. Since the gradient in the configuration space is complex, we replace the gradient smoother by a random perturbation smoother. The oracle smoother jointly calls the random perturbation smoother and a segment smoother over multiple iterations. These two smoothers are described in Algorithm 3 and 4.

**Algorithm 2:** GNNSmoother

**Input:** step size $\epsilon$, stop difference $\delta$, outer loop $L$, inner loop $K$
**Input:** explored path $\pi : (v_i, v_i')_{i \in [0,k]}$, free samples $V_f$, collided samples $V_c$
**for** $n \in \{1 \ldots L\}$ **do**
   $G \leftarrow \{V : \{V_\pi, V_f, V_c\}, E : \text{k-NN}(V_\pi, V) \cup E_\pi\}$
   $\pi' : (u_i, u_i')_{i \in [0,k]} = \mathcal{N}_S(V, E)$
   **for** $m \in \{1 \ldots K\}$ **do**
      $d \leftarrow 0$
      **for** $u_i \in V_{\pi'}, i \in [1, k]$ **do**
         $w_i \leftarrow$ steer $v_i$ toward $u_i$ within step $\epsilon$
         **if** $e_{i-1} : (v_{i-1}, w_i) \subseteq C_{free}$ **then**
            Replace $v_i \in \pi$ with $w_i$
            $d \leftarrow d + ||w_i - u_i||_2^2$
         **end if**
      **end for**
      **if** $d \leq \delta$ **then**
         **break**
      **end if**
   **end for**
**end for**
**return** $\pi$

---

**Algorithm 3:** RandomSmoother

**Input:** path $\pi : (v_i, v_i')_{i \in [0,k]}$, perturbation range $\epsilon$, iteration $L_R$
**for** $n \in \{1 \ldots L_R\}$ **do**
   pick a random node $v_i \in \pi, 1 \leq i \leq k$
   $u_i \leftarrow v_i + \text{random}(-\epsilon, \epsilon)$
   **if** $(v_{i-1}, u_i), (u_i, v_{i+1}) \subseteq C_{free}$ and
   $Cost[(v_{i-1}, u_i), (u_i, v_{i+1})] < Cost[(v_{i-1}, v_i), (v_i, v_{i+1})]$ **then**
      replace $v_i$ with $u_i$
   **end if**
**end for**
**return** $\pi$

---

**Algorithm 4:** SegmentSmoother

**Input:** perturbed path $\pi : (v_i, v_i')_{i \in [0,k]}$ from Algorithm 3
$critical = [v_0, v_k']$
**for** $i \in [1, k]$ **do**
   **if** $(v_{i-1}, v_i') \not\subseteq C_{free}$ **then**
      append $v_i$ to $critical$
   **end if**
**end for**
$\pi_M = \emptyset$
**for** adjacent pair $v_i, v_j \in critical$ **do**
   $V \leftarrow \{v_p \mid v_p \in \pi, i \leq p \leq j\}$
   $E \leftarrow \{(v_a, v_b) \mid v_a, v_b \in V, (v_a, v_b) \subseteq C_{free}\}$
   $\pi_{ij} \leftarrow$ the shortest path from $v_i$ to $v_j$ via Dijkstra$(V, E)$
   $\pi_M \leftarrow \pi_M \cup \pi_{ij}$
**end for**
**return** $\pi_M$