# OpenReview forum: "Reducing Collision Checking for Sampling-Based Motion Planning Using Graph Neural Networks"
_NeurIPS.cc/2021/Conference — NeurIPS 2021 Poster_

### Official Review · Reviewer_FpjY · 2021-07-14

**Rating:** 6
**Confidence:** 3

**Summary:**

At a high-level, this work proposes GNNs as models for biasing the sampling of edges and nodes to connect in a random geometric graph. The chief goal is to reduce the amount of collision-checking necessary to plan a path from a start to a goal. Two models are presented: one that performs the above, and another that then smooths the returned path in order to further reduce path cost.

Specifically, the proposed approach first constructs a random geometric graph (RGG) by sampling points from free space. Then, a GNN-parameterized policy iteratively guides the selection of RGG edges for collision-checking. A collision-free path is found when the start and goal nodes are connected on the RGG. The algorithm then leverages another GNN to smooth the solution path. Both GNNs are learned by imitating the solutions of existing oracles. The proposed approach achieves higher success rates, lower collision checks, and comparable path costs on a set of benchmark motion planning tasks compared to existing traditional and learning-/sampling-based algorithms.

**Limitations And Societal Impact:**

There are some questions raised in the main review which apply as limitations.

Regarding societal impact, however, the Broader Impact section (Line 375 and onwards) does not really address the broader societal impact of advancements in motion planning. Motion planning is a fundamental capability of robotics in general, and so any advancements benefit both positive and negative societal impacts. The authors did mention safety, biases, and interpretability, although these topics should probably have received more of a discussion (especially so for motion planning where a robot is basing its motion around other humans, for example, based on a neural network output).

**Main Review:**

Strengths:

- This paper tackles a problem with high importance, focusing on collision-checking (a critical component in sampling-based motion planning algorithms) and how it can be improved (i.e., reducing how many times it is necessary) by learning-based techniques.

- Using GNNs to smooth discrete paths appears to this reviewer to be a novel technique.

- The theoretic probabilistic completeness of the underlying sampling-based motion planning algorithm is maintained even when combined with the GNN-based learning components.

- The experiments demonstrate that the proposed method is capable of reducing collision checks without compromising much in terms of solution quality.

Major Weaknesses:
- The writing has typos and is unpolished/imprecise in some areas. The paper would benefit from another editing pass. Some examples are:
    - Line 2 - "Checking collision" vs "Collision-checking" or "Checking collisions"
    - Line 20 - It is slightly unclear what "with a few degrees of freedom" means (it is also imprecise). Does it mean a small amount of degrees of freedom? If so, a slightly more precise way of writing this is "with few degrees of freedom"? Even better, stating how many degrees of freedom (roughly, is it 2-5, 10-20, 100?)
    - Line 71 - "which could be naturally incorporated with reinforcement learning or imitation learning." What does "incorporated" mean here? Would a better phrase be "naturally tackled with" or "where reinforcement learning or imitation learning could naturally be applied"?
    - Line 107 - "specifically focus" should be "specifically focuses"
    - Line 142 - "a typical attention function", there are many types of attention, and a citation should probably be present here referring to which exact type is being used. Also, the writing does not explicitly indicate that this is the exact attention mechanism used in this work.
    - Line 198 - Is the L2 distance actually what is appended? The L2 distance is usually indicated like $\parallel v - v_g \parallel_2$ rather than parentheses.
    - This is not an exhaustive list. In general, impreciseness makes it hard for readers to grasp the concepts being discussed.

- Motion planning has a rich set of prior literature and, as a result, it is difficult to capture all of the relevant work in the field in a paper with space constraints. However, related work such as (Mandalika et al. in ICAPS 2019) should be included and briefly discussed as they also addresses the issue of reducing edge evaluations in a large graph.

- The text on some of the figures (e.g., 2, 3, 5, and 7) is hard to read because of its small size.

- The rationale behind the design choices of the proposed algorithm are not explained very clearly. A lot of algorithm components are merely presented as-is. There is no insight or intuition as to why things are being done this way, what it enables, or how it addresses the problem. Some examples:
    - Line 198, why are these features chosen specifically? What is it about the L2 distance between a node and the goal that is more informative than the difference between them? "serve as heuristics for the GNN to be more informed about which node is more valuable" does not provide much information. Further, why does $y_l$ not use the same L2 distance feature?
    - How are obstacles encoded? Line 203 seems to indicate that they are boxes and the obstacle information is the extremal vertices? Is that correct?
    - Why does the method use a GNN smoother for post-processing? Why not use the gradient-informed path smoothing described in Section 5.2 instead? The experiments mainly show that smoothing in general is helpful for reducing path cost (although at the cost of increasing collision checks, sometimes significantly so), and it is difficult to identify if it is this _specific_ GNN-based smoother that is necessary or if another method such as the gradient-informed path smoothing described in Section 5.2 or simple rewiring (as in RRT*) would work just as well with the GNN-based path explorer.

- There are also a few questions about the experiments. Chiefly,
    - Why does RRT* fail about 20-40% of the time in higher-dimensional settings? Is it timing out?
    - Why does RRT* have way more collision checks in 2D compared to NEXT but way less in high-dim environments?

Minor Comments:

- Figure 5 (d) should probably use a logarithmic y-axis scale, since the slow NEXT values dominate the graph.
- Related to the comment about writing above, typos should also be removed such as the space between * and ( at the end of line 111.

**Time Spent Reviewing:**

7

---

> ### Author Response · Authors · 2021-08-10
> **Response to Reviewer FpjY**
>
> Thank you for your careful reading. We address the specific questions as follows.
>
> > The writing has typos and is unpolished/imprecise in some areas. The paper would benefit from another editing pass… rather than parentheses.
>
> We will improve the listed typos and the typo mentioned in minor comments in the updated paper.
>
>
> > Motion planning has a rich set of prior literature and, as a result, it is difficult to capture all of the relevant work in the field in a paper with space constraints. However, related work such as (Mandalika et al. in ICAPS 2019) should be included and briefly discussed as they also address the issue of reducing edge evaluations in a large graph.
>
> For a high-level clarification, please check our response to Reviewer fGnh. We will also add similar discussion with the lazy methods in the updated paper, including LazyPRM, LazySP, LWA*, LRA*, GLS, and StrOLL [lazy_prm, lazy_sp, lwa, lra, gls, lazy_experience].
>
> > The text on some of the figures (e.g., 2, 3, 5, and 7) is hard to read because of its small size.
>
> We will improve the font size, and also modify Figure 5(d) to be logarithmic scale in the updated paper.
>
>
> > Line 198, why are these features chosen specifically? What is it about the L2 distance between a node and the goal that is more informative than the difference between them? "serve as heuristics for the GNN to be more informed about which node is more valuable" does not provide much information. Further, why does yl not use the same L2 distance feature?
>
> The difference between the node and the goal simply implies which direction is most promising to reach the goal. On the other hand, when there are no feasible nodes to explore along that specific direction, we want the GNN to instead explore those nodes which are feasible but also provide relatively high-quality solutions. As a result, the L2 distance along each dimension serves as an ellipsoidal heuristic, informing which nodes have higher potentials even if they are not in the shortest direction [informed]. We do not use the same L2 distance feature for $y$, to make the formulation consistent with the update function in Equation 2. We will also add ablation study on the specific feature choices to the updated Appendix.
>
> > How are obstacles encoded? Line 203 seems to indicate that they are boxes and the obstacle information is the extremal vertices? Is that correct?
>
> The representations for the obstacles in our experiment are implemented as boxes. The obstacle information is not the extremal vertices, but the center and length of the obstacles along each dimension. We also provide these descriptions in Section 1.1 of the Appendix.
>
> > Why does the method use a GNN smoother for post-processing? Why not use the gradient-informed path smoothing described in Section 5.2 instead? The experiments mainly show that smoothing in general is helpful for reducing path cost (although at the cost of increasing collision checks, sometimes significantly so), and it is difficult to identify if it is this specific GNN-based smoother that is necessary or if another method such as the gradient-informed path smoothing described in Section 5.2 or simple rewiring (as in RRT*) would work just as well with the GNN-based path explorer.
>
> We provide a comparison between the GNN smoother and the oracle smoother, in Section 4.2 of the Appendix. As shown in Figure 3 of the Appendix, while maintaining a comparative quality of the path improvement, our GNN smoother takes much lower computational cost than the oracle smoother, in terms of both the collision checks and the total time.
>
> > Why does RRT* fail about 20-40% of the time in higher-dimensional settings? Is it timing out?
>
> For the failure cases, RRT* reaches the maximum budget of the sampling nodes and is forced to be terminated while not finding a solution.
>
> > Why does RRT* have way more collision checks in 2D compared to NEXT but way less in high-dim environments?
>
> There are two reasons why this happens. For the 2D environment, NEXT has way fewer collision checks, since it finds solutions with fewer samples, which shows its strength in simple environments.
>
> However, for high-dim environments, RRT* has way fewer collision checks, which is mainly benefited from its steer function. When expanding a new node, NEXT will predict the new node using its neural network, and connect directly to that node, while checking the collision of the samples along the edge. On the other hand, RRT* samples a node from free space, and determines a new node to connect to through the steer function. The steer function only generates nodes that are close to the search tree within a desired distance. Now NEXT requires more collision checks than RRT* for each tree expansion, since it is being too optimistic to make sure whether a node which is far away could directly connect to the current tree.
>
> References:
>
> [lazy_prm] Bohlin, R., & Kavraki, L. E. (2000, April). Path planning using lazy PRM. In Proceedings 2000 ICRA. Millennium Conference. IEEE International Conference on Robotics and Automation. Symposia Proceedings (Cat. No. 00CH37065) (Vol. 1, pp. 521-528). IEEE.
>
> [lazy_sp] Haghtalab, N., Mackenzie, S., Procaccia, A., Salzman, O., & Srinivasa, S. (2018, June). The provable virtue of laziness in motion planning. In Proceedings of the International Conference on Automated Planning and Scheduling (Vol. 28, No. 1).
>
> [lwa] Cohen, B., Phillips, M., & Likhachev, M. (2015, May). Planning single-arm manipulations with n-arm robots. In Eighth Annual Symposium on Combinatorial Search.
>
> [lra] Mandalika, A., Salzman, O., & Srinivasa, S. (2018, June). Lazy receding horizon A* for efficient path planning in graphs with expensive-to-evaluate edges. In Proceedings of the International Conference on Automated Planning and Scheduling (Vol. 28, No. 1).
>
> [gls] Mandalika, A., Choudhury, S., Salzman, O., & Srinivasa, S. (2019, July). Generalized lazy search for robot motion planning: Interleaving search and edge evaluation via event-based toggles. In Proceedings of the International Conference on Automated Planning and Scheduling (Vol. 29, pp. 745-753).
>
> [lazy_experience] Bhardwaj, M., Choudhury, S., Boots, B., & Srinivasa, S. (2019). Leveraging experience in lazy search. arXiv preprint arXiv:1907.07238.
>
> [informed] Gammell, J. D., Srinivasa, S. S., & Barfoot, T. D. (2014, September). Informed RRT*: Optimal sampling-based path planning focused via direct sampling of an admissible ellipsoidal heuristic. In 2014 IEEE/RSJ International Conference on Intelligent Robots and Systems (pp. 2997-3004). IEEE.

---

> > ### Comment · Reviewer_FpjY · 2021-08-31
> > **Response to Authors**
> >
> > Thank you very much for these comments, as well as the informative responses to the other reviewers.

---

> > > ### Author Response · Authors · 2021-08-31
> > > **Thank you**
> > >
> > > We thank the reviewer for the comments and for reading our response. We will be sure to update the paper according to the suggestions.

---

### Official Review · Reviewer_vokm · 2021-07-16

**Rating:** 7
**Confidence:** 4

**Summary:**

This paper proposes a learning-based algorithm for reducing the planning time for collision-free robot motion planning problems. The author rightly notes that the bottleneck for motion planning is collision checking, especially along the vertices on the edges of a search tree, and proposes to learn a function, called GNN Path Explorer, that prioritizes the edges to check collisions. Based on this priority information, the algorithm builds a path from a start to a goal, while checking collisions only on the edges that have the highest priority out of the ones that have not been considered so far. This tree is then fed to a GNN Path Smoother, which processes the path and modifies it to one that has a lower cost. Training data is generated by past planning experience, in which the oracle is used to compute the optimal path on the current search tree. It is demonstrated on several domains, ranging from 2D environments with narrow passages and a point robot, to domains with high dimensional configuration spaces (c-space).


**Limitations And Societal Impact:**

Yes

**Main Review:**

The paper has a clear motivation that stems from a correct understanding of the bottleneck in motion planning. I agree that the main shortcoming of the previous works, as the author notes, is the fact that they focus on reducing sample complexity rather than collision checking.

The technical aspect of the paper is sound. The idea of using the graph of vertices sampled from the c-space and edges constructed by the k-nearest-neighbor algorithm is much more plausible than the past works that use the workspace as inputs, and the use of GNN to process such input is appropriate. I also found the interaction between the Path Explorer and the planner to be interesting, novel, and useful. Unlike most algorithms that try to either learn a sampler or directly predict a path, iteratively adding edges from the RRG and to the tree not only speeds up planning but also preserves probabilistic completeness guarantee.

The experiments are convincing. The proposed algorithm reduces the number of collisions while improving the path cost in a variety of domains.

I have a couple of suggestions to improve the paper. The first is regarding the visualizations of the empirical results. The use of line-connected plots in Figure 5 (a) and (d) is inappropriate, as there is no trend across the x-axis that the figure is trying to illustrate. A bar graph seems more appropriate here. Also, instead of Figure 5(d), I would like to see a bar graph that shows the breakdown of the total planning time - how much it is due to collision checking, GNN prediction, etc. Lastly, the confidence intervals on Figures 5(b) and 5(c) would be needed.

The second is on the description of technical details. In 4.1, the paper states that GNN Path Explorer finds a feasible path (L157), but this is not true, and the GNN simply outputs the priority on the edges. The feasible path is built based on this priority prediction. I also had to read the paper a couple of times to understand the interaction between the RGG <V, E> and the tree T_i that it builds (I still don’t understand what E_i is). The description is rather scattered between sections 4.1 and 5.1, and the paper would benefit from pseudocode that describes the interaction.

I also have a couple of questions:

- Why does the Path Explorer use only V, E, and O? It seems it could also benefit from T_i as well
- Why do you only sample edges from the frontier of the tree? Isn’t there a chance of finding a solution by sampling edges to the internal node of the tree? Perhaps I misunderstood the term frontier - it typically refers to the leaf nodes of a tree. Here, it is referring to the edges. Can you define what frontier means?



**Time Spent Reviewing:**

Did not count

---

> ### Author Response · Authors · 2021-08-10
> **Response to Reviewer vokm**
>
> Thank you for your careful reading of our paper. We are glad you found our method to be a useful and novel solution to an important problem. We address the specific questions as follows.
>
> > I have a couple of suggestions to improve the paper... Lastly, the confidence intervals on Figures 5(b) and 5(c) would be needed.
>
> We will improve Figure 5(a) and 5(d) to be bar graphs in our updated paper, and also add the confidence intervals on Figure 5(b) and 5(c). Figure 5(d) will also include the breakdown of the planning time.
>
> > The second is on the description of technical details...the paper would benefit from pseudocode that describes the interaction.
>
> $E_i$ means the edges that connect one of the nodes on the current tree $\mathcal{T}_i,$ to an unexplored node (see Line 224-225). We provide the pseudocodes of all our algorithms in Section 6 of the Appendix.
>
>
> > Why does the Path Explorer use only $V,E,$ and $O$? It seems it could also benefit from $\mathcal{T}_i$ as well.
>
> There are two perspectives that the Path Explorer only uses $V,E,$ and $O$. From the design perspective, the priority of an edge indicates both the cost and the feasibility for the path starting from this specific edge to the goal, which does not involve the current search tree $\mathcal{T}_i$. Such design is also beneficial from a practical perspective, because we only need to feed the whole graph to the GNN once to get the priorities, which saves the computation time.
>
> > Why do you only sample edges from the frontier of the tree? Isn’t there a chance of finding a solution by sampling edges to the internal node of the tree? Perhaps I misunderstood the term frontier - it typically refers to the leaf nodes of a tree. Here, it is referring to the edges. Can you define what frontier means?
>
> The frontier means the edges that connect one of the nodes on the current tree $\mathcal{T}_i,$ to an unexplored node (see Line 224-225). By such definition, these edges on the frontier are also able to connect to the internal node of the tree.

---

### Official Review · Reviewer_fGnh · 2021-07-18

**Rating:** 7
**Confidence:** 5

**Summary:**

The paper leverages graph based learning to improve sampling based motion planning, specifically on reducing edge evaluations and also on smoothing computed paths for reducing path length.

**Limitations And Societal Impact:**

Yes

**Main Review:**

The paper's idea of using graph NN for finding patterns from an initial sample of nodes in the workspace is very interesting. The key motivation for the paper to pursue this work is to address the expensive collision checking operations by reducing the number of evaluations. However, the paper neglects the existing field of lazy motion planning which addresses the same problem as this paper[lazy_prm,lazy_sp, gls] . Methods have shown to use much lower number of edge evaluations to find the path~[gls] and also have leveraged learning to improve lazy planning [lazy_experience]. For e.g. [lazy_experience] can solve the 7d manipulator problem within 35 edge evaluations while the proposed method takes 350 evaluations (though on different environments]. The proposed method needs to evaluated with existing lazy motion planning methods to show if the proposed method will benefit the community.

Learning based motion planning also enables using sensory perception directly without requiring models of the environment. If the proposed method cannot compare to lazy motion planning methods, showing the method working with vision based collision avoidance~[object_collision] could open up some application areas for the approach and be a useful contribution.




References:

[lazy_prm] Bohlin, Robert, and Lydia E. Kavraki. "Path planning using lazy PRM." Proceedings 2000 ICRA. Millennium Conference. IEEE International Conference on Robotics and Automation. Symposia Proceedings (Cat. No. 00CH37065). Vol. 1. IEEE, 2000.

[lazy_sp] Haghtalab, N., Mackenzie, S., Procaccia, A., Salzman, O., & Srinivasa, S. (2018, June). The provable virtue of laziness in motion planning. In Proceedings of the International Conference on Automated Planning and Scheduling (Vol. 28, No. 1).

[gls] Mandalika, A., Choudhury, S., Salzman, O., & Srinivasa, S. (2019, July). Generalized lazy search for robot motion planning: Interleaving search and edge evaluation via event-based toggles. In Proceedings of the International Conference on Automated Planning and Scheduling (Vol. 29, pp. 745-753).

[lazy_experience] Bhardwaj, M., Choudhury, S., Boots, B., & Srinivasa, S. (2019). Leveraging experience in lazy search. arXiv preprint arXiv:1907.07238.

[object_collision] Danielczuk, M., Mousavian, A., Eppner, C., & Fox, D. (2020). Object rearrangement using learned implicit collision functions. arXiv preprint arXiv:2011.10726.

**Time Spent Reviewing:**

3

---

> ### Author Response · Authors · 2021-08-10
> **Response to Reviewer fGnh**
>
> Thank you for your important suggestion for the comparison with lazy search methods, as we were mainly comparing GNN against existing sampling-based approaches with deep learning components. With the same goal of reducing collision checking, our methods and lazy search are very different in the following ways.
>
> > … The proposed method needs to be evaluated with existing lazy motion planning methods to show if the proposed method will benefit the community.
>
> Lazy search reduces collision-checking by using a greedy heuristic: it only evaluates edges along shortest paths or subpaths (without considering collision) from start to goal, updates the graph if any in-collision edges are found, and then replans the shortest path/subpath in the updated graph and repeats the process [lazy_prm, lazy_sp, gls, lazy_experience]. Therefore, it inherits the downside of greedy search strategies: the shortest path in a graph without considering collision can often be very far from the actual optimal paths, which need to avoid the obstacles in roundabout ways. Consider the typical benchmark case of a U-shaped obstacle, with the start state inside and the goal outside, both close to the bottom of the U. Without considering the U-shaped obstacle, the shortest path simply crosses the bottom of the U, which is the initial path in a lazy search strategy. It is easy to see that it takes a long sequence of collision checking and graph update for the lazy search to find the path that escapes the U-shape. The learning-based approach in [gls] follows the same lazy search strategy of the iterative search of shortest subpaths and uses experience to accelerate the updates, and the overall approach is still biased by the greedy strategy. As a result, lazy search behaves well when the estimated shortest paths happen to have few collisions -- take the 7D manipulator example that the reviewer cited, if a collision-free path is only 32 edge collision-checks away from the shortest path without considering collision, then lazy search can perform very well. However, that is certainly not the typical case in challenging or random problems, where the actual collision-free paths are usually far from the geometric shortest path which makes the planning problems challenging in the first place.
>
> The GNN-based methods that we propose avoid the dependency on greedy heuristics or any fixed strategy, but use environment-specific training to determine the priority of edge exploration. The GNN explorer only suggests one-step lookahead at a time locally (Figure 2), based on patterns that are learned from training data, rather than being biased by the global geometric shortest path. Take the same U-shaped obstacle as an example again: The GNN explorer can recognize the graph structure, and propose edges that directly point to the opening side of the U-shaped obstacle. Although lazy-search can be proved to be edge-optimal [lazy_sp] in the family of planners that **do not receive additional information**, the benefit of using learning-based components is exactly to use additional information that can be gained from training, and thus cannot be compared in this same class of planners.
>
> Following the reviewers’ suggestion, we have implemented lazy search as another baseline method to compare with. A snapshot of the results are as follows, and we will update the paper with the complete tables. As we see in the first table, over 1000 random instances, the GNN approach uses fewer collision checking than lazy search (LazySP) on average, and the gap widens on high-dimensional problems (14D: 486.95 vs 789.25). In fact, over these random instances it is easy to find many environments where lazy search requires over 200% of the collision checking needed by GNN. We should also note that the training set can be further adapted to focus on problems that the fixed greedy strategy in lazy search is particularly bad at. Moreover, as shown in the second table, the frequent replanning needed in lazy search resulted in much longer runtime (over 500% in high-dimensional problems) compared to GNN-based methods. This is a main challenge of the lazy-based method because it needs to update the graph repeatedly to maintain the global estimation on shortest paths, whereas our method avoids it by learning an adaptive priority. We believe these results show the value of GNN-based approaches proposed in our work and we will further elaborate them in the paper.
>
> | Collision Checking | Easy2D | Hard2D | Kuka7D | 14D |
> | :---: | :--- |:--- |:--- |:--- |
> |        GNN        | 336.25±3.71 | 715.65±6.76  | **350.52±8.29**  | **486.95±12.99** |
> |      **LazySP**       | 351.80±2.47 | 801.21±6.74  | 369.36±19.42 | 604.64±38.84 |
> |  GNN + Smoother   | 496.79±4.68 | 1029.72±8.33 | 477.32±9.06  | 791.78±14.27 |
> |    GNN w/o OE     | **332.30±4.00** | **703.72±5.78**  | 353.89±7.47  | 547.16±37.61 |
> |   GNN w/o OE + Smoother   | 565.38±6.37 | 1126.02±9.91 | 466.06±6.70  | 789.25±36.54 |
>
> | Total Running Time  |   Easy2D   |    Hard2D    |    Kuka7D    |     14D      |
> | :---: | :--- |:--- |:--- |:--- |
> |        GNN         | **35.07±0.78** |  80.57±1.16  |  **56.77±1.89**  |  **84.25±3.16**  |
> |       **LazySP**   | 83.65±1.97 | 287.30±11.93 | 339.97±61.63 | 465.30±93.14 |
> |   GNN + Smoother   | 48.71±0.79 |  99.40±1.18  |  72.32±1.93  | 119.68±2.95  |
> |     GNN w/o OE     | 35.43±0.66 |  **80.19±1.04**  |  57.62±1.87  |  95.19±8.36  |
> |    GNN w/o OE +  Smoother  | 51.59±0.76 | 101.40±1.17  |  72.39±1.85  | 125.09±8.25  |
>
> We wish to emphasize that the contribution of our paper is not to show “learning is all we need” but to show that a particular choice of GNN-representations and the corresponding training methods are important to improve existing learning-based methods. The proposed methods can be directly used in a lazy search framework as well, or together with other classical heuristics, and the GNN models will make the learning components work better.

---

> > ### Comment · Reviewer_fGnh · 2021-08-19
> > **Thanks for the analysis with Lazy planning**
> >
> > Thanks for the detailed explanation and comparison to lazy planning. This was very helpful.

---

> > > ### Author Response · Authors · 2021-08-19
> > > **Thank you**
> > >
> > > We thank the reviewer for the important question and for reading our explanation. We will be sure to update the paper with the discussion and comparison.

---

### Decision · Program_Chairs · 2021-09-27

**Decision:**

Accept (Poster)

**Comment:**

The paper seeks to improve the computational efficiency of sample-based motion planning by reducing the number of collision checks. The paper proposes an algorithm that learns to prioritize the order in which edges in a sample-based planning graph are checked for collision. Following this prioritization, a path is generated between the start and goal nodes and then smoothed to reduce cost. Results on a variety of different planning domains demonstrate that the proposed approach is more computationally efficient than contemporary methods, while still providing paths with similar cost.

The paper considers an important problem in sample-based robot motion planning, namely reducing the computational cost associated with collision checking. The reviewers agree that the work is well motivated and that the proposal to use graph neural networks to guide edge evaluations is technically sound, novel, and principled. The experiments effectively convey the advantages of this strategy. The reviewers raised some initial concerns, notably those related to the merits relative to lazy motion planning strategies, most if not all of which were adequately addressed in the author response and discussion phase. The authors are encouraged to make sure that the next revision of the paper reflects this discussion.